# Trade-off between tree planting and wetland conservation in China

Yi Xi [1], Shushi Peng [1✉], Gang Liu[1], Agnès Ducharne [2], Philippe Ciais [3,4], Catherine Prigent[5,6], Xinyu Li[1] & Xutao Tang[1]

Trade-offs between tree planting programs and wetland conservation are unclear. Here, we employ satellite-derived inundation data and a process-based land surface model (ORCHI-DEE-Hillslope) to investigate the impacts of tree planting on wetland dynamics in China for 2000–2016 and the potential impacts of near-term tree planting activities for 2017–2035. We find that 160,000–190,000 km$^2$ (25.3–25.6%) of historical tree planting over wetland grid cells has resulted in 1,300–1,500 km$^2$ (0.3–0.4%) net wetland loss. Compared to moist southern regions, the dry northern and western regions show a much higher sensitivity of wetland reduction to tree planting. With most protected wetlands in China located in the drier northern and western basins, continuing tree planting scenarios are projected to lead to a > 10% wetland loss relative to 2000 across 4–8 out of 38 national wetland nature reserves. Our work shows how spatial optimization can help the balance of tree planting and wetland conservation targets.

[1] Sino-French Institute for Earth System Science, College of Urban and Environmental Sciences, and Laboratory for Earth Surface Processes, Peking University, Beijing, China. [2] Sorbonne Université, CNRS, EPHE, Laboratoire METIS (Milieux environnementaux, transferts et interaction dans les hydrosystèmes et les sols), 75005 Paris, France. [3] Laboratoire des Sciences du Climat et de l'Environnement, LSCE/IPSL, CEA-CNRS-UVSQ, Université Paris-Saclay, 91191 Gif-sur-Yvette, France. [4] The Cyprus Institute, 20 Konstantinou Kavafi Street, 2121 Nicosia, Cyprus. [5] CNRS, Sorbonne Université, Observatoire de Paris, Université PSL, LERMA, Paris, France. [6] Estellus, Paris, France. ✉email: speng@pku.edu.cn

A fforestation and reforestation have been proposed as effective, safe, and affordable natural climate solutions to lock up carbon and mitigate climate change[1]. Since the 1990s, afforestation has been widely implemented by many countries, especially China[2]. In response to a national priority of protecting ecological services and land-system sustainability in the context of rapid economic development, China implemented a series of large-scale afforestation and forest protection programs such as the Three-North Shelterbelt Program[3], the Natural Forest Conservation Program[4], and the Grain for Green Program[5] during the last four decades[6]. According to China's ninth National Forest Inventory (NFI) covering the period 2014–2018, the total forest area in China increased by ~50% (~0.6 million km[2], hereafter Mkm[2]) relative to the 1980s; this increase was predominant by plantation forest (Fig. 1a, b). Whilst the unprecedented increase of forest area in China has successfully reduced soil erosion, dust storms, desertification, and improved flood mitigation[7–10], the large-scale afforestation has also increased evapotranspiration (ET) and reduced runoff (Q) and soil moisture (SM), especially in drylands of northern China[11–16]. At the catchment scale, this reduction in available water inevitably reduces water delivered to wetlands, thereby posing a threat to the wetlands' wide range of ecosystem services from food and water security to climate regulation and their cultural and spiritual

importance[17]. Furthermore, continuing tree planting in China could also jeopardize the conservation of natural wetlands, a critical component in the achievement of the Sustainable Development Goals (SDGs) under the United Nations' Agenda 2030, and thereby induce trade-offs with afforestation for climate SDGs[17,18]. However, despite China's forest area has increased from 1.4 Mkm[2] to 1.8 Mkm[2] (+26%) during 2000–2018 (Fig. 1b), where and to what extent tree planting threatens wetland conservation is not yet clear. A new ambitious tree planting plan under which China would expand its total forest coverage to 26% of the country by 2035 (Fig. 1b) is presented in the national 15-year Comprehensive Plan for Ecological System Protection and Recovery Work, released in June 2020 (ref. [19]). As a consequence, gaining an understanding of all the hydrological and ecological consequences of China's tree planting programs is quite urgent if we are to evaluate and negotiate the trade-off between tree planting and wetland conservation.

To quantify the impacts of China's past and future tree planting programs on wetland areas, we combine satellite-based inundation data from the Global Inundation Estimate from Multiple Satellites version 2, hereafter GIEMS-2 (ref. [20]), and a state-of-the-art land surface model, ORCHIDEE-Hillslope (Organizing Carbon and Hydrology In Dynamic Ecosystems–Hillslope)[21,22]. This model partitions the water in each model grid cell, accounting for the

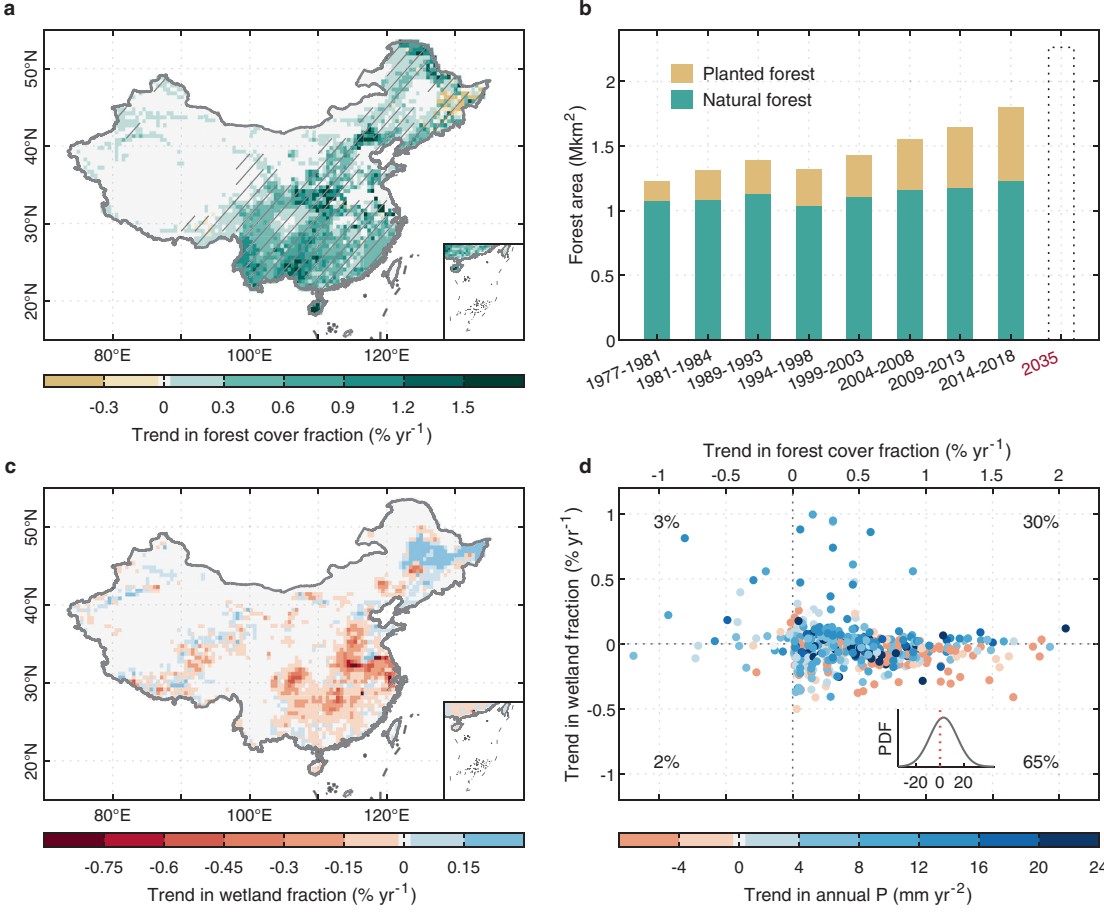

**Fig. 1 Historical change of forest coverage and wetland extent in China. a** Spatial pattern of trend in forest cover fraction from 2000 to 2016 from forest inventory data. **b** Temporal change in a forest area during the last four decades compiled from China's 2nd–9th National Forest Inventory. **c** Spatial pattern of trend in wetland fraction from 2000 to 2015 from GIEMS-2. **d** Trend in wetland fraction versus trend in forest cover fraction from 2000 to 2015 across grid cells (*n* = 562) in (**a**), (**c**). The color of each point shows the trend in annual precipitation (P) from GSWP3-W5E5. The inset at the bottom right of Fig. (**d**) indicates the probability density function of the trend in annual P across points in the fourth quadrant. The trends in Fig. (**a**), (**c**) are estimated by linear least-squares regression and hatching indicates the trend is statistically significant (*t*-test, *p* < 0.05). Please note that the grid cells with a >10% rice paddy coverage are removed using the HYDE v3.2 data set to avoid wetland change induced by human activities in **d** panel.

wetter lowlands which result from the convergence of hillslope water flows (see details in Methods). Based on the SM output from ORCHIDEE-Hillslope, the flooded area dynamics of each grid cell is then calculated using a subgrid hydrological model[23–25] calibrated to match satellite observations of inundated areas (Methods). We first perform historical simulations for 2000–2016 with ORCHIDEE-Hillslope forced by climate data from GSWP3-W5E5 (refs. [26,27]) and annual land-cover maps based on forest change from China's NFI. The contribution of forest change to wetland change is isolated with two factorial simulations: S0 (simulation with climate change (hereafter called CC), elevated $CO_2$ concentration (hereafter $eCO_2$), but without forest change) and S1 (simulation with CC, $eCO_2$, and forest change). Further, to evaluate the effects of China's near-term tree planting on future wetland areas, we perform similar factorial simulations for 2017–2035 but using land-cover maps following the near-term tree planting plan (Methods). Our work demonstrates that the wetlands located in dry climate zones are more vulnerable to tree planting than those in wet climate zones, and reminds us of a reasonable spatial optimization of future tree planting activities for the trade-off between the carbon sequestration from forest gains and wetland conservation targets in China.

## Results

**Historical wetland change in response to afforestation.** Figure 1a shows where large-scale tree planting programs have been implemented in China. These programs led to a substantial increase in forest coverage since 2000 across almost all regions. In northern, southwestern, and central China (see the map of China's nine regions in Supplementary Fig. 1), forest coverage increased at a rate above 0.3% per year from 2000 to 2016 over an average of $0.5° \times 0.5°$ grid of our model (Fig. 1a). Concurrently, the satellite-based global inundation product GIEMS-2 (ref. [20]) reports a nonsignificant loss in the inundated areas (including open water, rice paddies, and wetlands) varying from −0.003% to −0.4% per year across ~73% of wetland grid cells (hereafter the wetland grid cells are defined as cells with a mean annual maximum wetland fraction >1%), in particular in northern, central, and southern China (Fig. 1c). Northeastern China is the only region where a wetland gain is diagnosed by GIEMS-2, and where the forest coverage has been reduced by about −0.3% $yr^{-1}$ during the last two decades. Excluding regions with flooding for rice cultivation (Methods), we found that ~68% of wetland grid cells across China show opposite trends between forest and wetland areas from 2000 to 2015. Among those, more than 95% experienced a coincident increase in forest area and a decreasing wetland extent, despite ~60% of them having had increasing annual precipitation (P) (Fig. 1d). Such opposing trends for forest and wetland areas even in regions experiencing increased precipitation, suggest a possible negative impact of tree planting on wetland areas in China, despite the human disturbances including wetland drainage, irrigation expansion, or other water infrastructure projects could also have negative impacts[28–30].

To quantify the impacts of afforestation on water delivery to wetlands, we first use a conceptual water-balance model, the Budyko framework[31] (Methods). This method was applied in a previous study to estimate the hydrological legacy of deforestation on global wetlands[32,33]. By assuming negligible changes in soil and groundwater storage over the annual cycle, the Budyko model expresses the partitioning of annual water supply (i.e., P) to ET or Q (runoff) within catchments according to the ratio of the potential demand for atmospheric water (i.e., potential evapotranspiration, PET) to annual P (Methods; Eqs. (2) and (3)). With lower water retention ability (i.e., a higher Q/P) for catchments planted with grasses than those planted with trees, the conversion from grasses to forests leads to increased

ET and decreased Q (Supplementary Fig. 2). Using the equation employed by Woodward et al. (ref. [33]) (Methods), we estimate that annual Q decreased by 0.8–2.4 mm $yr^{-2}$ at grid scale and by 0.15–0.9 mm $yr^{-2}$ at catchment scale from 2000 to 2016 across regions with substantially increasing forest area (Fig. 2a and Supplementary Fig. 3a). More forest plantations lead to larger reductions in Q (Fig. 2b and Supplementary Fig. 3b). When normalizing the whole decrease of Q ($\delta Q$) throughout the period by annual P ($\delta Q/P$), the stronger impact of afforestation on Q is found in mesic and dry regions with a mean annual PET/P (Supplementary Fig. 4) of 0.6–2.2, where the decrease in Q due to 20% forest gains is equivalent to more than 2% of annual P (Fig. 2b). These results imply different hydrological consequences of afforestation in different climate zones. In line with this conceptual framework, similar spatial patterns of decreasing Q due to forest gains during the study period (mirroring increasing ET) were also simulated by the ORCHIDEE-Hillslope, but with a slightly lower magnitude (−0.4 to −1.6 mm $yr^{-2}$; Fig. 2c, e). The simulated relationship between $\delta Q/P$ and PET/P by ORCHIDEE-Hillslope thus follows the Budyko conceptual model well (Fig. 2d and Supplementary Fig. 3d). Moreover, the process-based model simulates a significantly decreasing trend in annual mean SM ($<−0.15$ mm $yr^{-2}$, $p < 0.05$) due to afforestation, especially in northeastern, northern, and southwestern China (Fig. 2f).

To further investigate the impacts of afforestation on wetland areas, we combine the mean SM estimates from ORCHIDEE-Hillslope with TOPMODEL, a subgrid hydrology model that redistributes the water table according to heterogeneous topographic conditions, to diagnose the subgrid fraction occupied by wetlands (flooded areas) (see details in Methods). TOPMODEL and its variants have been widely applied to diagnose grid-scale saturated fractions on the basis of high-resolution topography distribution[24,25,34,35]. After calibration with two satellite-based wetland products, the Regularly Flooded Wetlands map (RFW)[36] and GIEMS-2 (ref. [20]) (see wetland definition in Methods), our wetland model can reproduce the observed spatial patterns and temporal variations of wetland extent (Supplementary Figs. 5–6; hereafter the results calibrated with RFW are shown in the main text, and results with GIEMS-2 in Supplementary). From 2000 to 2016, out of a total of 0.33 $Mkm^2$ of afforestation across the nation, 0.19 $Mkm^2$ of trees were planted in wetland grid cells, which resulted in 1500 $km^2$ net wetland loss in China (−0.3% of total wetland area). Despite the net relative wetland loss attributed to afforestation from our simulations being modest at the country scale, in northern and northeastern China, the wetland loss trend has been more dramatic, with loss rates larger than 0.04% $yr^{-1}$ (~1 $km^2$ $yr^{-1}$ in one grid with an area of ~2500 $km^2$; Fig. 3a). In contrast, fewer grid cells show significant wetland loss due to afforestation in southern China, although the forest increase is equal to or even larger in this region (Fig. 1a). This suggests that the impact of afforestation per $km^2$ on wetland loss is enhanced in dry compared to wet climate zones.

**Sensitivity of wetland change to afforestation.** Figure 3b shows the sensitivity of the change in wetland area ($A_{wet}$) to the change in forest area ($A_{forest}$), i.e., the wetland loss per one $km^2$ afforestation. The regions with higher sensitivity are concentrated in northern and northeastern China, overlapping with the areas where we inferred substantial wetland change due to forest gains (Fig. 3a). During the last two decades, every $km^2$ increment of forest area in these regions can be interpreted as a 0.005–0.10 $km^2$ loss of wetland extent (Fig. 3b). In contrast, southern China experienced a substantial and extensive forest gain, but the sensitivity is no larger than 0.005 $km^2$ $km^{-2}$ in most areas. To gain a better understanding of the response of wetlands to afforestation, we decompose the marginal wetland sensitivity to forest change ($\frac{\delta A_{wet}}{\delta A_{forest}}$) into the sensitivity of wetland change to soil moisture

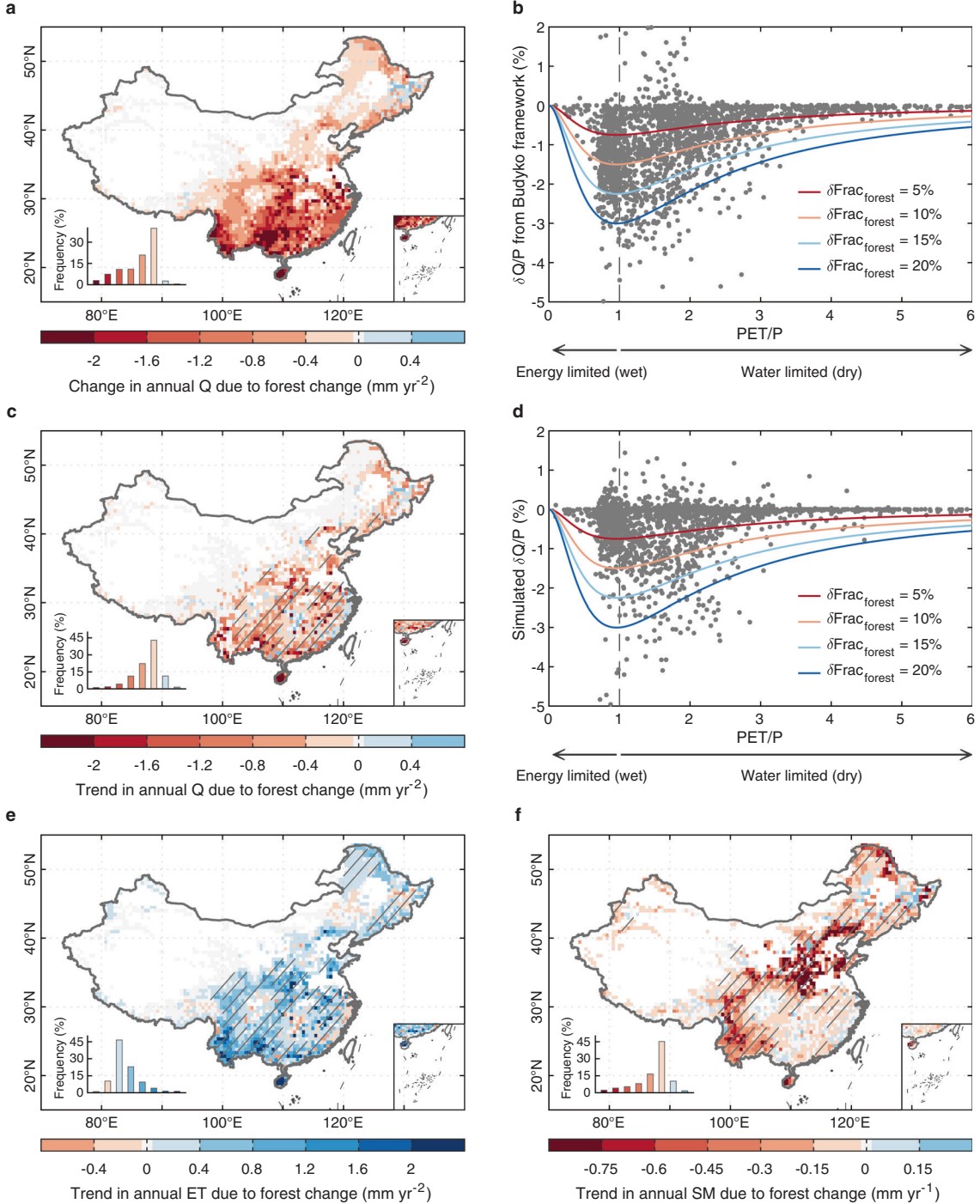

**Fig. 2 Change in the hydrological cycle in response to historical forest change in China. a** Change in annual runoff (δQ) due to forest change from 2000 to 2016 according to the Budyko framework. **b** Relationship between the ratio of δQ to precipitation (P) from the Budyko framework and the ratio of potential evapotranspiration (PET) to P (PET/P) under different levels of forest gains (5, 10, 15, and 20%). The dark gray dots (n = 2203) represent the δQ/P due to the forest change at the grid scale, calculated as the Budyko framework. **c, e, f** Spatial patterns of simulated trends in Q, ET, and soil moisture (SM) due to forest change from ORCHIDEE-Hillslope. **d** Same as Fig. (**b**), but with the δQ/P simulated from ORCHIDEE-Hillslope (n = 2203). The trends in Fig. (**c**), (**e**), and (**f**) are estimated by linear least-squares regression and hatching indicates the trend is statistically significant (t-test, p < 0.05).

$\left(\frac{\delta A_{\text{wet}}}{\delta \text{SM}}, \text{km}^2 \text{ wetlands per mm SM}\right)$ following TOPMODEL and the sensitivity of soil moisture to forest area change $\left(\frac{\delta \text{SM}}{\delta A_{\text{forest}}}, \text{mm SM per km}^2 \text{ forest area}\right)$ using the identity of Eq. (1). The results are shown in Fig. 3c–e and Supplementary Fig. 7.

$$\frac{\delta A_{\text{wet}}}{\delta A_{\text{forest}}} = \frac{\delta A_{\text{wet}}}{\delta \text{SM}} \times \frac{\delta \text{SM}}{\delta A_{\text{forest}}} \quad (1)$$

$\delta A_{\text{wet}}$, $\delta A_{\text{forest}}$, and $\delta \text{SM}$ represent the change in wetland area, forest area, and SM from 2000 to 2016, respectively. Across the three climate zones classified by PET/P (Supplementary Fig. 4), $\frac{\delta A_{\text{wet}}}{\delta A_{\text{forest}}}$ rises gradually from wet regions (PET/P <1; ~0 km² km⁻²) to dry regions (PET/P >2; ~ −0.014 km² km⁻²) (Fig. 3c), and this pattern is mainly controlled by the $\frac{\delta \text{SM}}{\delta A_{\text{forest}}}$ (Fig. 3e). Specifically, in wet southern areas where the abundant precipitation maintains a

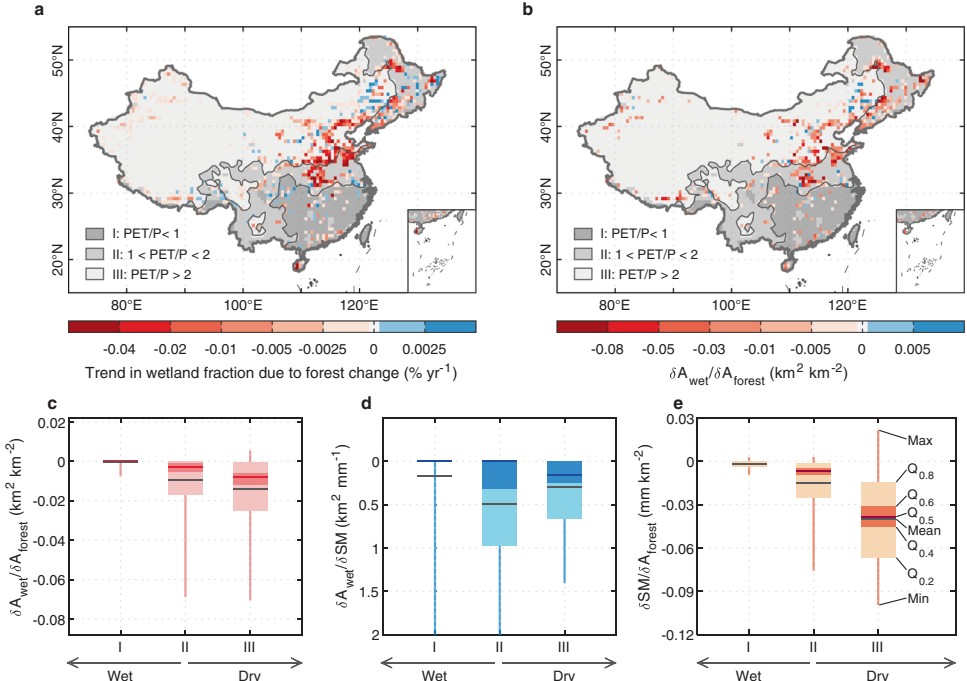

**Fig. 3 Historical wetland change in response to forest change in China. a, b** Spatial patterns of wetland change due to the forest change (**a**) and its sensitivity to forest change (**b**) from 2000 to 2016, with the average annual ratio of potential evapotranspiration (PET) to precipitation (P), (PET/P) as background. The trends in Fig. (**a**), (**b**) are estimated by linear least-squares regression and hatching indicates the trend is statistically significant (*t*-test, $p < 0.05$). **c–e** Sensitivity of wetland change to forest change ($\delta A_{wet}/\delta A_{forest}$) (**c**), wetland change to soil moisture (SM) change ($\delta A_{wet}/\delta SM$) (**d**), and SM change to forest change ($\delta SM/\delta A_{forest}$) (**e**), at grid scale for three climate zones divided by PET/P as shown in Fig. **a**. Please note that in order to clearly show the statistical results of three sensitivities, the grid cells with no forest from 2000 to 2016, or no change in forest cover fraction, or a less than 1% wetland fraction from RFW are not considered, and the outliers outside of the range of 5th–95th percentiles are dropped in Fig. **b–e**. This results in $n = 322$, $n = 317$, and $n = 243$ for three climate zones, respectively in Fig. **c–e**. The whiskers and boxes in Fig. **c–e** indicate the maximum (Max), 80% ($Q_{0.8}$), 60% ($Q_{0.6}$), 50% ($Q_{0.5}$), 40% ($Q_{0.4}$), 20% ($Q_{0.2}$), minimum (Min), and mean (Mean) sensitivity of all grid cells in each climate zone after dropping outliers.

moist soil throughout the year, the negligible influence of afforestation on SM results in little response of wetland area to forest change, even with a substantial $\frac{\delta A_{wet}}{\delta SM}$ in some grid cells (Fig. 3d). On the contrary, in dry regions where more than 80% of annual precipitation is lost to the atmosphere as ET (Supplementary Fig. 2a), the substantial increase of ET due to afforestation leads to a more negative $\frac{\delta SM}{\delta A_{forest}}$ ($\sim -0.04$ mm km$^{-2}$) and therefore a higher $\frac{\delta A_{wet}}{\delta A_{forest}}$. In the mesic climate zone with a PET/P of 1–2, $\frac{\delta A_{wet}}{\delta SM}$ is larger than in wet or dry regions (Fig. 3d), which thus raises the $\frac{\delta A_{wet}}{\delta A_{forest}}$ in this region.

According to the List of Protected Wetlands in China[37], we identify the basins, at level 6 as classified by the global HydroBASINS database[38] containing National, Provincial, as well as Municipal and County-level wetland conservations (hereafter called BAS$_N$, BAS$_P$, and BAS$_{MC}$, respectively; Fig. 4a) and then investigate the potential impacts of continuous tree planting on those protected basins in China. Lying in northeastern, northern, and central China (Fig. 4a), the protected basins cover ~35% of the country (~3.31 Mkm$^2$) but account for ~43% (~0.69 Mkm$^2$) of the national forest area and ~46% (~0.30 Mkm$^2$) of wetland extent from RFW (~52% in GIEMS-2), and hold importance for conserving rare waterbirds, plants, and water resources (Supplementary Fig. 8). During the last two decades, in response to the ~0.08 Mkm$^2$ forest increase in these basins (~24% of national forest increase), we infer that ~800 km$^2$ wetlands (~53% of national wetland loss) disappeared across these basins, which have a ~30% more negative $\frac{\delta A_{wet}}{\delta A_{forest}}$ than the national average and a ~70% more negative value than the unprotected basins (Fig. 4b). Among

different levels of wetland conservation, the highest-level BAS$_N$ presents a higher risk of wetland loss from afforestation ($-0.018$ km$^2$ km$^{-2}$) against BAS$_P$ and BAS$_{MC}$ ($-0.010$ and $-0.009$ km$^2$ km$^{-2}$ for BAS$_P$ and BAS$_{MC}$, respectively), relating to their broader spatial coverage in the dry climate zones with higher $\frac{\delta A_{wet}}{\delta A_{forest}}$. If the historical forest increase of BAS$_N$ and all protected basins were reduced by ~10 km$^2$ (~0.003% of the national forest increase) and ~40 km$^2$ (~0.01%), respectively, 2 out of 39 BAS$_N$ (~5%) and 8 out of 144 protected basins (~6%) would be saved from a >10% wetland loss (Supplementary Fig. 9a, b).

**Near-future wetland change due to forest change.** As a natural climate solution[1], large-scale afforestation and forest conservation is a keystone of the strategy of China to incept a net-zero greenhouse gas emission pathway[19,39]. Although historical wetland change due to forest change has been limited thus far (Supplementary Fig. 9), expanding afforestation in the future could lead to an important shift in the wetland areas. Following the national 15-year ecological plan, we assume a near-term scenario for 2017–2035 with linearly increasing forest area and constant climate as in 2000–2016 (hereafter S$_A$) to investigate the wetland loss due to near-term tree planting in China (Supplementary Table 1; Methods). Relative to historical tree planting activities, the forest gains in S$_A$ follow the historical trajectory, primarily occurring in southern and northeastern China (Supplementary Fig. 10a). By 2035, we project that a total of ~1300 km$^2$ of wetland will be lost in response to 0.22 Mkm$^2$ of new tree planting in wetland grid cells (Supplementary Fig. 11b), and 11% of wetland grid cells are projected to have a >10%

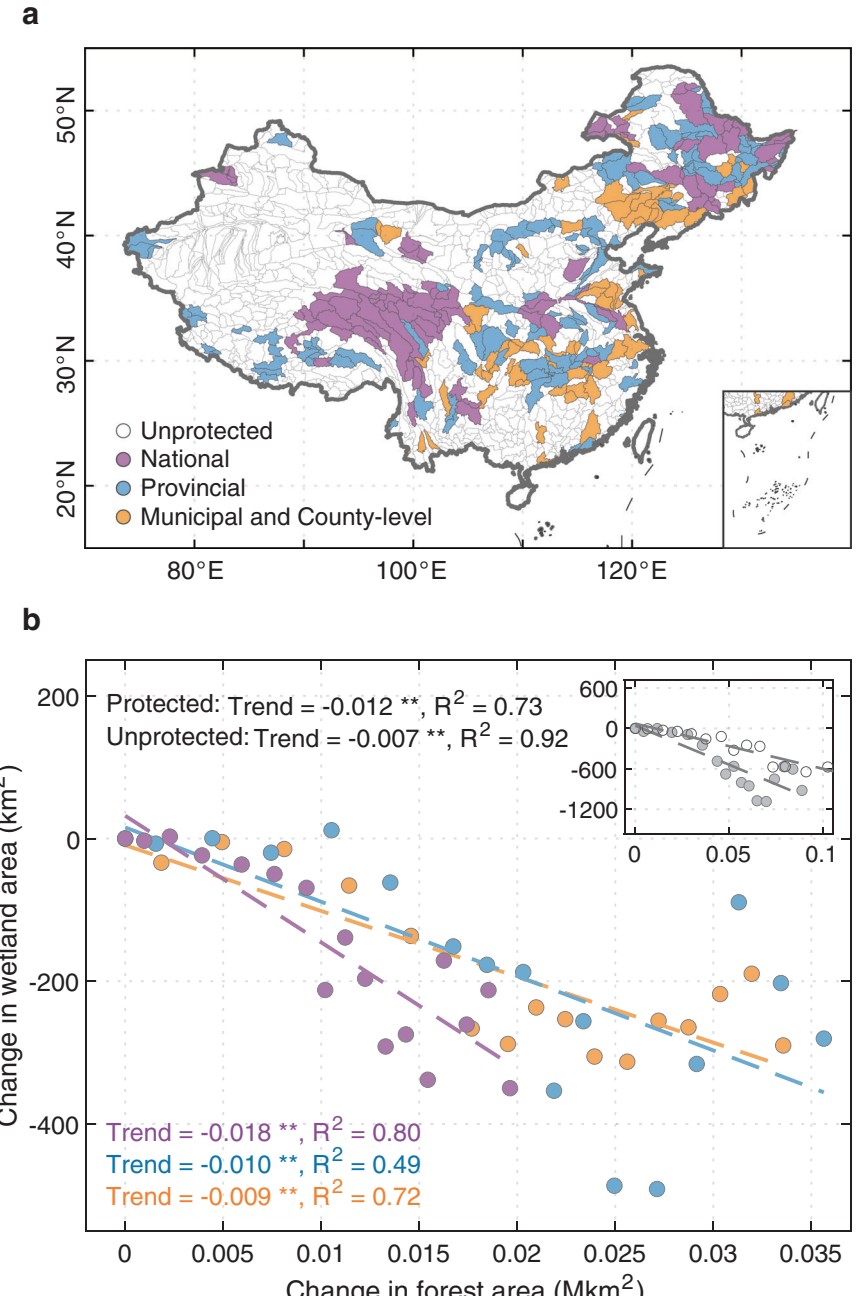

**Fig. 4 Historical wetland change due to forest change across protected basins. a** Spatial distributions of basins containing unprotected (transparent), national (purple), provincial (blue), and municipal and county-level (orange) wetland nature reserves. **b** Change in wetland area versus change in forest area across basins with different wetland conservation and all protected and unprotected basins (the inset) from 2000 to 2016. Two asterisks indicate statistical significance at the 99% confidence level. Note that since the forest area is monotonically increasing, the points in Fig. **b** from left to right correspond to the years from 2000 to 2016.

wetland loss relative to 2000 due to tree planting (Fig. 5). Although the projected near-term wetland loss still accounts for no more than 1% of baseline wetland area, the effects of afforestation on wetland conservation are concentrated in the dry regions with high $\frac{\delta A_{wet}}{\delta A_{forest}}$, implying that special attention is needed in choosing the catchments for future afforestation.

To evaluate the effects of different afforestation locations on China's wetland conservation, we design three additional extreme scenarios for $S_A$ to plant all new trees into the drier climate zone where PET/P >2 ($S_A^{dry}$), or into the mesic climate zone with PET/P between 1 and 2 ($S_A^{mesic}$), or into the wet climate zone of PET/P <1

($S_A^{wet}$) (Fig. 5b–d and Supplementary Fig. 11c–e). From 2017 to 2035, these three scenarios are projected to experience 0.16, 0.18, and 0.32 Mkm² of forest area increase across grid cells containing wetlands, leading to 1800, 1600, and 200 km² of net wetland loss, respectively (Supplementary Fig. 11). Specifically, under the $S_A^{dry}$ scenario, due to the highest $\frac{\delta A_{wet}}{\delta A_{forest}}$ in the drier climate zone where PET/P >2, the smallest afforestation area across wetlands grid cells would result in the largest wetland loss and the most deteriorated wetland grid cells compared to the other two climate zones (Fig. 5). The total wetland loss under this scenario is projected to be ~1.5 times more than in the control scenario. The

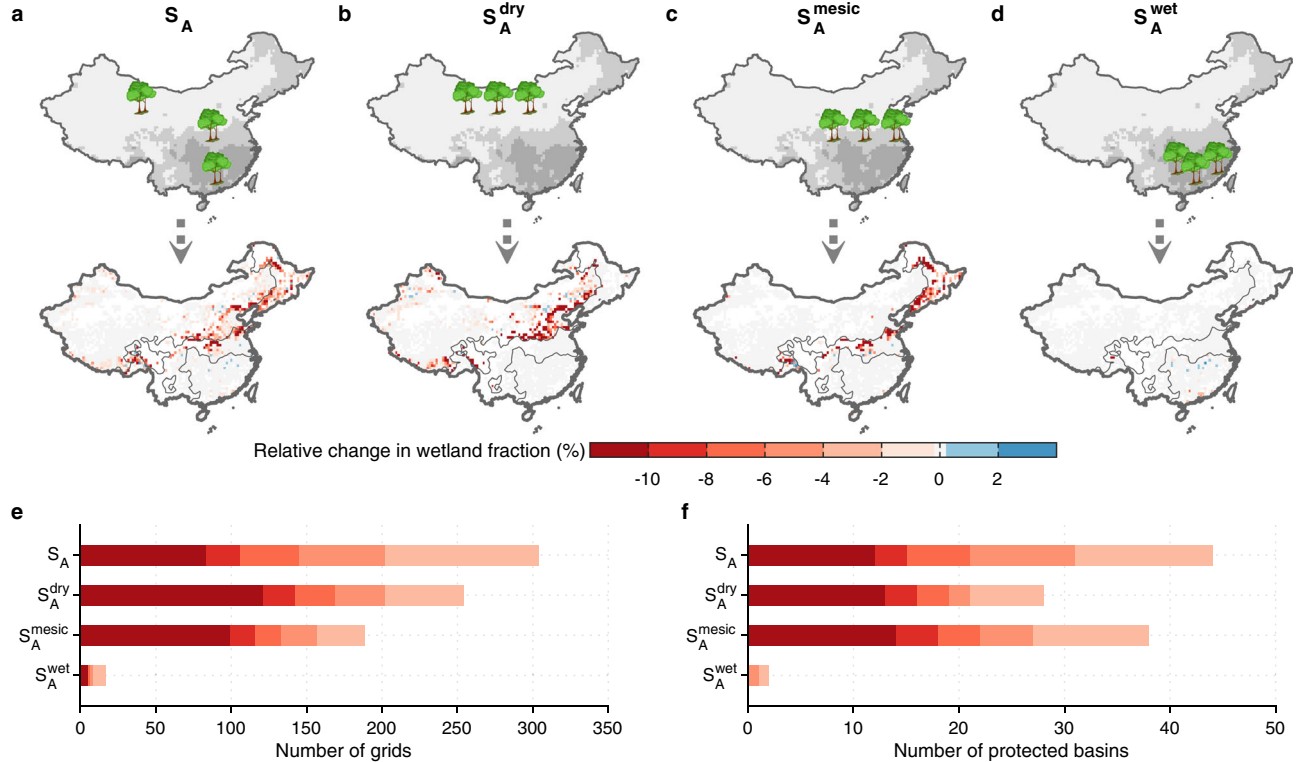

**Fig. 5 Near-term wetland change due to forest change in China. a–d** Four different tree-planting scenarios for 2017–2035 and corresponding consequences on wetlands relative to the baseline wetland extent in 2016. **e, f** Numbers of grid cells and protected basins with a 2–4, 4–6, 6–8, 8–10%, and >10% wetland loss relative to 2016, respectively under four near-term scenarios. The gray shading corresponds to the climate zone classified by the ratio of potential evapotranspiration (PET) to precipitation (P), (PET/P) from Fig. 3.

scenario $S_A^{mesic}$ has a similar wetland degradation consequence to $S_A^{dry}$, related to the high $\frac{\delta A_{wet}}{\delta SM}$ in the intermediate climate zone of PET/P in the range 1–2. In contrast, owing to the low $\frac{\delta A_{wet}}{\delta A_{forest}}$, even with twice the forest increase than in $S_A^{dry}$ and $S_A^{mesic}$, the extreme expansion of forest area in wet areas of $S_A^{wet}$ has little wetland loss. Hence, under the same target of the national afforestation area, the choice of afforestation locations has very different consequences on wetland conservation.

Annual precipitation in China is projected to experience a small change (−11 to 12 mm yr$^{-1}$, <2%) by 2035 under three shared socioeconomic pathways (SSP1-2.6, SSP3-7.0, and SSP5-8.5) from the multi-models ensemble of the Inter-Sectoral Impact Model Intercomparison Project 3b (ISIMIP3b)[40] (Supplementary Figs. 12, 13). The total wetland area simulated with ORCHIDEE-Hillslope forced by near-term climate forcing from ISIMIP3b and identical land-cover maps with $S_A$ (hereafter $S_B$), therefore, shows an insignificant trend of −300 to 80 km$^2$ yr$^{-1}$ ($p > 0.1$) from 2017 to 2035 (Fig. 6a). Considerable disagreements in projected wetland change can be found due to the highly uncertain precipitation simulated by different climate models under different SSP scenarios (Supplementary Table 2). By contrast, continuous tree planting following the national 15-year ecological plan consistently leads to a significant net wetland loss (1200–1300 km$^2$, $p < 0.001$) in China by 2035 across SSP1-2.6, SSP3-7.0, and SSP5-8.5 (Fig. 6b), accounting for about 23, −92, and 41% of wetland change from multi-model mean projections, respectively (Supplementary Table 2). With the higher $\frac{\delta A_{wet}}{\delta A_{forest}}$ in dry northern China, projected forest gains show a higher contribution in the two climate zones with PET/P >1 (−41 to 215%) than PET/P <1 (13–18%), in BAS$_N$ and BAS$_P$ (−47 to 514%) than BAS$_{MC}$ (13–27%) (Supplementary Table 2). These findings suggest that

the wetland loss induced by near-term tree planting activities cannot be offset by the subtle precipitation change under the three scenarios of future climate change.

## Discussion

Over the next few decades, China will continue to implement a series of large-scale afforestation initiatives to protect ecological services and land-system sustainability and combat climate change[19,39]. A key prerequisite is to evaluate the consequences of historical tree planting and its interaction with other ecological services[41,42]. Here our study shows that the unprecedented increase of forest area in China from 2000 to 2016 only leads to a 1300–1500 km$^2$ (0.3–0.4%) net wetland loss, however, a detailed sensitivity analysis suggests that the wetlands are more vulnerable to forest increase in the dry climate zones of northern and northeastern China. Most of the protected wetlands in China are distributed across dry northern China, therefore suffering a higher risk of wetland loss during the period 2000–2016. According to the near-term tree planting plan in China, we show that if tree planting follows the historical trajectory it will lead to an additional 1300 km$^2$ wetland loss by 2035, all concentrated in the dry regions. To lower the risk of wetland loss due to forest change, planting in areas with low wetland sensitivity to forest increase or far away from wetlands could avoid the risk of wetland deterioration. Serving as a warning, our findings remind us that a reasonable spatial optimization of future tree planting activities could help to balance the carbon sequestration from forest gains and the protection of precious wetland resources in China as well as other arid and semi-arid regions in the world such as the western United States, Central Asia, and Central Africa.

One limitation of our analysis is the lack of considering the land-atmosphere feedback of afforestation. A previous study[16]

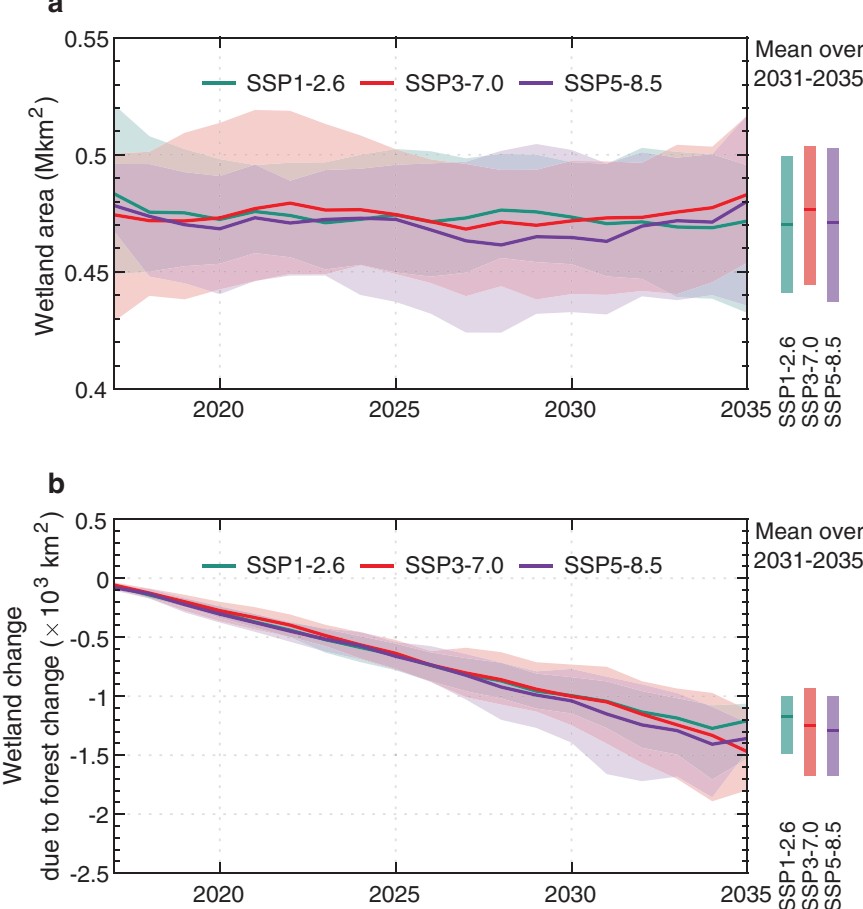

**Fig. 6 Time series of wetland change due to near-term climate change and forest change in China. a** Near-term wetland change under $S_B$ scenario (including climate change, elevated atmospheric $CO_2$ concentration, and forest change) under three shared socioeconomic pathways (SSPs) for 2017–2035 in China. **b** Near-term wetland change due to forest change under three SSPs for 2017–2035 in China. The solid lines show the multi-model mean projections (5-year moving average), and the shading presents the likely ranges estimated from the five individual climate models. The time means for 2031–2035 are shown as colored vertical bars.

using the coupled land-atmosphere global climate model shows that continuous afforestation in China from 1982 to 2011 has increased annual precipitation in southeastern China (20 mm yr⁻¹ decade⁻¹, $p < 0.1$) and northern China (8 mm yr⁻¹ decade⁻¹, $p > 0.1$). In this way, the simulated wetland loss due to afforestation in northern China could be alleviated by the feedback on precipitation. Moreover, given the large uncertainties in the simulated hydrological cycle by the coupled climate models[16,43], we appeal to more coupled simulations to investigate the integrated effects of afforestation on wetlands. As more and more attention is being paid to the benefits of carbon sequestration from afforestation[1,44,45], if and where afforestation can alleviate wetland loss by increasing local and/or downwind precipitation is of great significance for all countries or regions experiencing substantial forest gains[46].

Many paired watershed studies have revealed that the effects of tree planting on water yield vary across species, forest age, and types of forest management[47,48]. These factors should be taken into account to further disentangle the trade-off between tree planting and wetland conservation, which are not fully represented in the current land surface model. Furthermore, the trade-off between tree planting and wetland conservation cannot be seen as only the superposition of tree planting and its hydrological consequences. The ecological trade-offs between wetlands and forests should also resolve the effectiveness of forest gains for carbon sequestration against wetlands[49], biophysical climate

feedbacks such as decreasing albedo due to forest increase in boreal regions[50], and other ecosystem services such as biodiversity conservation and cultural values[17]. The implementation of future tree planting activities should also face the aspect of the feasibility of spatial optimization of afforestation goals and the priority of ecosystem services[49,51,52].

## Methods

**Forest inventory data**. To investigate the forest change in China during the last few decades, we collected data from the second to ninth National Forest Inventory (NFI) released by China's State Forestry Administration (data for the first NFI are not available)[53,54]. At the province scale, the forest inventory in China is carried out every 4–5 years, covering 1973–2018. Forests in the NFI are defined as lands with more than 20% tree cover, including arbor forests, bamboo forests, and shrubs. Bamboo forests and shrubs are under-represented in ORCHIDEE. We, therefore, only focus on arbor forests in this study; they account for ~90% of all forest area and ~80% of the annual increase in forest area in China. To generate a forest map for each year, we combined the spatial information from the 1:1,000,000 Chinese Vegetation Map[55] with the annual forest area change linearly interpolated from the five-year-interval forest inventory data. The annual forest area changes at the province level from the forest inventory data were proportionally allocated to the forest grid cells at 0.5° × 0.5° spatial resolution in the province. More details about the algorithm can be found in Xi et al. (ref. [56]) and Li et al. (ref. [16]). To validate the spatial distributions and temporal variations in forest cover fractions from our inventory-based forest maps, we used two satellite-based data sets of forest cover fractions from Moderate Resolution Imaging Spectroradiometer (MODIS) and Song et al. (ref. [57]) (Supplementary Table 3). Detailed comparisons of spatial patterns and temporal change in forest coverage show that the inventory-based data can match well with satellite-based forest data (Supplementary Text 1).

**Observation-based wetland extent (flooded area)**. We used two observation-based wetland data sources to conduct wetland-related analysis. First, to estimate historical wetland change, we used the satellite-based global inundation product GIEMS-2 (Global Inundation Estimate from Multiple Satellites version 2; ref. [20]) to estimate the wetland change in China from 2000–2015. By combining passive and active microwaves, along with visible and near-infrared observations, this $0.25° × 0.25°$ product gives monthly estimates of surface water extent, including wetlands, open water, and rice paddies, but potentially excluding large lakes, rivers, and reservoirs. Some small water bodies that cover less than 10% of the grid could be missed in GIEMS-2 due to the relatively coarse resolution and the dense vegetation. The spatial distribution and temporal variation of the global inundation area from GIEMS-2 have been validated with existing independent products such as precipitation and altimeter river height[20]. Since we were interested in natural wetlands, we removed the inventory-corrected and dynamic rice areas from the HYDE v3.2 data set[58] from GIEMS-2 when analysing the impacts of tree planting on wetlands in Fig. 1d. Aggregated into $0.5° × 0.5°$, the long-term maximum and mean annual maximum of China's (global) inundated extent for the period 2000–2015 from GIEMS-2 are 1.8 (9.7) and 0.7 (4.1) $Mkm^2$ after removing the rice paddies.

Second, to calibrate the parameters of the wetland model, we used the satellite-based map of regularly flooded wetlands (RFW)[36]. RFW is a static, high-resolution (15 arc-sec) wetland map, generated by overlapping GIEMS-D15 (downscaled from GIEMS-1)[59], the ESA-CCI land-cover map[60], and global surface water bodies[61]. Thus, RFW potentially includes small wetlands missed by GIEMS-2 at a $0.25° × 0.25°$ resolution. Aggregated into $0.5° × 0.5°$, China's (global) inundated extent from RFW is 1.9 (11.8) $Mkm^2$ after removing the rice paddies, and it is regarded as a long-term maximum wetland extent in this study. Owing to the substantial uncertainty of the wetland maps, we also used the annual maximum wetland area from GIEMS-2 from 2000 to 2015 to calibrate the parameters of the model (Supplementary Figs. 14–17).

**Protected wetland locations in China**. To evaluate the impacts of forest change on basins with protected wetlands, we used the List of Protected Wetlands in China[37]. Established in the 1950s, by 2013 the List had grown to include 2622 nature reserves and 318 wetland nature reserves (WNRs). In total, these WNRs cover ~0.29 $Mkm^2$, mainly distributed across northeastern, northern, and central China and Qinghai-Tibetan Plateau. Due to the small landscape structure of most WNRs, we extracted the basins at level 6 as classified by the global HydroBASINS database[38] intersecting with these WNRs, to investigate the effects of forest change on protected wetlands at a basin scale.

**Budyko conceptual model**. The Budyko framework was proposed five decades ago. It is based on the empirical relationship between annual mean ET/P and PET/P (ref. [31]). By assuming changes in soil and groundwater storage are negligible over annual time scales, it indicates that most P in a catchment is allocated to Q in wet and energy-limited areas, while most P goes to ET in dry and water-limited areas (Supplementary Fig. 2). Despite the simple and "lumped parameter" structure, the intuitive framework has been successfully applied to explain and predict how the terrestrial hydrological cycle has changed up until now[33,62–64]. We employed the approach here following Woodward et al. (ref. [33]) to explain conceptually the change in the allocation of P after planting trees. Since we were interested in the effects of vegetation changes on ET here, we used the equation developed by Zhang et al. (ref. [32]):

$$\frac{ET}{P} = \frac{1 + w \times \frac{PET}{P}}{1 + w \times \frac{PET}{P} + \left(\frac{PET}{P}\right)^{-1}} \quad (2)$$

where $w$ is a coefficient related to water availability for plants. Typically, $w = 2$ for forest and $w = 0.5$ for grass owing to their different rooting depths[32]. According to the water-balance equation (P = ET + Q), the relationship between Q/P and PET/P can be expressed following Woodward et al. (ref. [33]) as:

$$\frac{Q}{P} = \frac{\left(\frac{PET}{P}\right)^{-1}}{1 + w \times \frac{PET}{P} + \left(\frac{PET}{P}\right)^{-1}} \quad (3)$$

Using the P and PET calculated from ORCHIDEE-Hillslope, we show the Budyko curves with different values of the parameter $w$ according to Eqs. (2) and (3) at grid and basin scale (Supplementary Fig. 2). After the conversion from grass to forests, ET tends to increase while Q is expected to decrease. The change of annual Q due to the forest cover change can be derived[33]:

$$\delta Q = P \times (f_t - f_{t-1}) \times \left[ \left( 1 - \frac{1 + w_f \times \frac{PET}{P}}{1 + w_f \times \frac{PET}{P} + \left(\frac{PET}{P}\right)^{-1}} \right) - \left( 1 - \frac{1 + w_g \times \frac{PET}{P}}{1 + w_g \times \frac{PET}{P} + \left(\frac{PET}{P}\right)^{-1}} \right) \right] \quad (4)$$

where $f_t$ and $f_{t-1}$ indicate the forest cover fraction at times $t$ and $t-1$, and $w_f$ and $w_g$ are the plant-available water coefficients for forest and grass respectively. The

spatial patterns of $\delta Q$ at grid and basin scale are shown in Fig. 2a and Supplementary Fig. 3a. According to this equation, a more intensive forest change means a more substantial change of Q, while the more obvious loss of Q ($\delta Q$) normalized by P ($\delta Q/P$) occurs in regions with a PET/P of 0.6–2.2, where the decrease in Q due to 20% forest gain is equivalent to more than 2% of P (Fig. 2).

**ORCHIDEE-Hillslope Simulation**. The ORCHIDEE land surface model[21] simulates the terrestrial carbon and hydrological processes and has been widely used for the detection and attribution of the global or regional carbon and hydrological cycles[65–67]. The carbon module simulates photosynthesis, litterfall, and soil carbon dynamics, while the hydrological module describes the partitioning of P into ET and Q, and the water redistribution in the 2-m soil column is based on the Richards equation[66,68,69]. Both the carbon and water processes in ORCHIDEE are very dependent on the vegetation cover, which is described as a mosaic of up to 13 plant functional types (PFT) including bare soil, nine forest types, C3 and C4 grasslands, and croplands for each grid cell. To prevent trees from accessing the SM required to grow grass and crops, the soil water budget is performed separately in three "soil tiles", one for the forest PFTs, one for grasslands and croplands, and one for bare soil. These soil tiles share the same P, but produce different surface runoff, infiltration, ET, and drainage at the bottom of the soil layer, as a result of different surface properties and soil moisture. Eventually, the sum of surface runoff and drainage from all soil tiles is transferred to the river system of the grid cell by means of two linear reservoirs, representing the lags of surface and subsurface flow, respectively. Each grid cell also includes a series of linear reservoirs representing the river. River discharge is then deduced from grid-cell to grid-cell routing along the river network[70].

ORCHIDEE-Hillslope (r6515) is based on the latest version of ORCHIDEE (tag 2.0), used in the IPSL-CM6-LR climate model[71] for the Climate Model Intercomparison Project Phase 6 (CMIP6), which was modified to describe the effects of hillslope hydrology on the subgrid-scale distribution of soil moisture and wetlands[22]. To this end, we introduced a new tile into each grid cell, representing the "lowland" part of the landscape, with a high propensity to be wet as it receives surface and subsurface flow generated in the upland part. This change, together with an impervious bottom at 2 m, allows a water table to build up, and feed baseflow to the river, as well as enhance ET compared to the upland fraction, where the 2-m soil is disconnected from the water table. For simplicity, the lowland fraction is constant over time in each grid cell and prescribed from RFW[22]. The land cover is assumed to be the same in the upland and lowland fraction, by lack of clear guiding rules to do otherwise[72]. Overall, at the grid-cell scale, ORCHIDEE-Hillslope leads to a higher SM, higher ET, but smaller Q compared to the standard version. An evaluation against independent observations in the Seine River basin showed that ORCHIDEE-Hillslope simulates a more realistic absolute value and seasonal cycle of river discharge and terrestrial water storage[22].

To evaluate the impacts of forest change on wetland change in China, we performed two sets of simulations using ORCHIDEE-Hillslope: with and without forest change (S1 and S0) for 2000–2016 (Historical scenarios) and for 2017–2035 (Near-term scenarios including $S_A$ and $S_B$) at $0.5° × 0.5°$ spatial resolution (Supplementary Table 1). The two historical simulations were conducted with the GSWP3-W5E5 climate forcing[26,27] and time-varying $CO_2$ concentrations from NOAA observations[73] for 2000–2016, but using different land-cover maps. The S1 uses land-cover maps for 2000–2016 generated from the forest inventory data from NFI, while S0 uses the constant land-cover map in 2000. They continue a single 200-year spinup simulation performed by repeating the climate forcing of a 20-year cycle (1980–1999) from GSWP3-W5E5, with constant $CO_2$ concentration (368 ppm as of 2000) and land-cover map in 2000. The four groups of near-term simulations (scenarios $S_A$ (planting trees following the national 15-year ecological plan), $S_A^{dry}$ (planting all trees under $S_A$ to the dry climate zone with PET/P >2), $S_A^{mesic}$ (planting all trees under $S_A$ to the mesic climate zone with PET/P ranging in 1–2), and $S_A^{wet}$ (planting all trees under $S_A$ to the wet climate zone with PET/P <1)) continue the historical ones, with the same 19-year climate forcing randomly generated from GSWP3-W5E5, constant $CO_2$ as in 2016, but four different tree-planting scenarios for S1 (Fig. 5a–d) and constant land-cover map in 2016 for S0. To investigate if near-term climate change will alleviate the wetland loss due to tree planting, we performed a similar scenario to $S_A$, called $S_B$, but using future climate forcing from ISIMIP3b (ref. [40]). The ISIMIP3b project includes three future shared socioeconomic pathways (SSPs), SSP1-2.6, SSP3-7.0, and SSP5-8.5, with five climate models (GFDL-ESM4, IPSL-CM6A-LR, MPI-ESM1-2-HR, MRI-ESM2-0, and UKESM1-0-LL) in each SSP. These near-future simulations continue the "Transient" simulations, which are similar to the Historical scenario but using historical climate forcing from climate models in ISIMIP3b. The spinup simulations were also performed for the "Transient" scenario. Please see Supplementary Table 1 for more details about simulated protocols and see Supplementary Text 2 for the algorithm to produce the annual land-cover maps.

**Simulation of wetland fraction**. To simulate the subgrid wetland extent and its dynamics, we used a TOPMODEL-based diagnostic model that has successfully predicted the spatial distribution and seasonality of natural wetlands extents[24,25,34]. Based on a few simplifying assumptions, the classical TOPMODEL offers an analytical relationship between SM deficit with respect to soil saturation and a topographic index[23,74]. It allows one to estimate the distribution of saturated areas, often regarded as wetlands, at the spatial resolution of the topographic information. To avoid numerous calculations from the input topography data, the initial

TOPMODEL framework has been simplified with some diagnostic algorithms, which directly link SM deficit and wetland fraction[24,34,75]. In this study, we used the algorithm of Stoker et al. (ref. [24]) as implemented in Xi et al. (ref. [25]). The monthly SM deficit is calculated from the SM output from ORCHIDEE-Hillslope. The key parameters of the diagnostic model are calibrated with the long-term maximum wetland extent from RFW[36] and annual maximum wetland area from GIEMS-2 for 2000–2015 (Supplementary Figs. 14–17). The comparison of simulated wetland extent with RFW and GIEMS-2 shows reasonable spatial patterns and time series of wetland extent in our simulations (Supplementary Figs. 5, 6).

**Reporting summary**. Further information on research design is available in the Nature Research Reporting Summary linked to this article.

## Data availability

All observation and model data that support the findings of this study are available as follows. The National Forest Inventory data were available from China's State Forestry Administration (http://www.forestry.gov.cn/). The GIEMS-2 data set analyzed during the current study over China from 2000 to 2015 has been deposited on Zenodo (https://doi.org/10.5281/zenodo.5750962)[76]. The RFW data sets are available at https://doi.pangaea.de/10.1594/PANGAEA.892657. The HYDE v3.2 data set are available at https://easy.dans.knaw.nl/ui/datasets/id/easy-dataset:74467. The historical and future climate data from GSWP3-W5E5 and ISIMIP3b are obtained from https://esg.pik-potsdam.de/search/isimip/. The protected wetland locations in China are obtained from http://www.zrbhq.cn/web/confirm.html. The shapefile data of basins at level 6 as classified by the global HydroBASINS database are available at https://www.hydrosheds.org/downloads.

## Code availability

The ORCHIDEE-Hillslope model (r6515) code used in this study is open-source and distributed under the CeCILL (CEA CNRS INRIA Logiciel Libre) license. It is available at https://forge.ipsl.jussieu.fr/orchidee/wiki/GroupActivities/CodeAvalaibilityPublication/ORCHIDEE-Hillslope-r6515, with guidance to install and run the model at https://forge.ipsl.jussieu.fr/orchidee/wiki/Documentation/UserGuide. All ancillary data to run the model can be accessed upon reasonable request to the corresponding author, with the response received in 1 week. The code to simulate the wetland area by TOPMODEL (Version v1.0) is publicly available on GitHub (https://github.com/yixixy/Wetland_simulation_by_TOPMODEL)[77]. The processing MATLAB codes are available at https://github.com/yixixy/Treeplanting_Wetlands_China.

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

## Acknowledgements

The study was supported by the National Natural Science Foundation of China (grant numbers 41830643 and 41722101), and the National Key Research and Development Program of China (2016YFC0500203). P.C. acknowledges support from the ANR CLAND convergence institute. We are grateful for the computational resources provided by the High-performance Computing Platform of Peking University's supercomputing facility.

## Author contributions

S.P. designed the study. Y.X. and A.D. performed model simulations. Y.X., G.L., and X.L. performed the analysis and Y.X. created all the figures. Y.X. and S.P. drafted the manuscript. Y.X., S.P., G.L., A.D., P.C., C.P., X.L., and X.T. contributed to commenting and writing on the draft manuscript.

## Competing interests

The authors declare no competing interests.
