## [Peer Review File · Nature Communications]

Reviewer #1 (Remarks to the Author):

The revised manuscript has improved substantially, and now discusses the trade-offs of afforestation and wetlands in a much more balanced and informative way. The authors have addressed the majority of my previous concerns. My only remaining concern that I hope the authors could address as well, is the role of alternative potentially covarying drivers to changes in wetland exchange. Otherwise, most of my comments below are intended to help the authors increase the clarity and nuance of their writing.

Beginning of Abstract and Conclusions. Afforestation programmes in China and elsewhere are motivated by multiple purposes beside climate mitigation, which the authors also acknowledge (e.g., at L32: "In response to a national priority of protecting ecological services and land-system sustainability in the context of rapid economic development, China implemented a series of large-scale afforestation and forest protection programmes"). Nevertheless, the starting sentences in the Abstract and Conclusions strongly frame the primary motivation (and the perceived benefit) of afforestation programmes as "climate mitigation", and leaves the impression that the studied trade-off is between "climate" and "ecological" objectives. Thus, perhaps consider deleting "to solve the climate change problem" or adding other reasons, for a more nuanced and accurate description of the objectives of the afforestation programmes in China in the same sentence. Drawing the readers' attention to reasons such as "prevention of desertification and land degradation" also might enhance the understanding of the challenges to spatial optimization. I am aware that the authors acknowledge all the climatic, ecological, social development targets associated with both afforestation and wetlands, but those statements are a little more hidden, whereas the first sentences in the Abstract and Conclusions come off very strongly.

L77 (i.e., section "Historical wetland change in response to afforestation"). While the other sections of the manuscript, better isolated the effect of afforestation, this section appears to claim to have isolated the "afforestation" effect (causation) despite essentially resting on the assumption that afforestation does not for example strongly co-vary with wetland drainage, irrigation expansion, or other water infrastructure projects.

In my previous review, I was concerned that "alternative drivers" to wetland change were not sufficiently accounted for. By "alternative drivers", I had in mind such land-use and water-use driven drivers (apologies that I did not articulate this even clearer in the previous exchange). For example, (Zhang et al. 2019) found "farm production, total aquatic products, and irrigated area" to be the top drivers of "wetland changes in the Beijing-Tianjin-Hebei Region". The authors might want to address this in some way, for example, by adding layer(s) of trends in irrigation, agricultural areas, or other proxies of agricultural activities, and inspect the covariation with forestation trends, and discuss the possibilities of such alternative explanations.

L241 (and Supplementary Table 2). I found it quite confusing that the contribution of forest change varied so greatly between ("accounting for about 23%, -92%, and 41% of wetland change from multi-model mean projections") and that the term "contribution" is defined in a way that makes >100% contributions possible. If I understand the Supplementary Table 2 and the authors' definitions correctly, the forest change effect on wetland extent is in fact quite consistent across the different scenarios, whereas the CC and eCO₂ effects can be erratic and both increase and decrease wetland extents. The authors might want to add an explanation of why CC and eCO₂ effects on wetland vary widely among the different scenarios. In addition, a different way to define "contribution", or a different term to describe the S1-to-S0 ratio, might be helpful for the reader.

Discussion/Conclusions. I appreciate the revised discussions and conclusions, which are much more nuanced. I find it a great addition that the authors added this at the end of the Conclusions:

"The ecological trade-offs between wetlands and plantations should face the aspect of other ecosystem services such as high carbon sequestration, biodiversity, and cultural values. The implementation of future afforestation activities should also assess the feasibility of spatial optimization of afforestation goals and the priority of ecosystem services." . The authors might want to add a few more sentences to explain what this means, and how challenging such trade-offs might or might not be to overcome. (For example, is climate mitigation through afforestation less efficient in the northern part less efficient due to albedo-effects anyway? Are the carbon stocks in wetlands large? Are the carbon stocks in wetlands well-accounted for? Etc.)

Supplementary Figure 11. Please update the in-figure subheadings S-A to S-D.

Terminology:

The term "afforestation" is normally reserved for forestation measures on historically non-forested land. In this manuscript, the term appears to be used to refer to both afforestation and reforestation. The authors might want to consider defining the term in the introduction to avoid confusion, or change the term to e.g., "forestation", "tree cover increase", or "afforestation and reforestation" etc.

Could the authors also add a definition of wetlands? Perhaps by expanding the definition at L 85 and explaining what a wetland fraction indicates? (I understand that lakes and rice paddies are removed, but how about other artificial reservoirs for example?)

References

Zhang, Liyun, Quan Zhen, Min Cheng, and Zhiyun Ouyang. 2019. "The Main Drivers of Wetland Changes in the Beijing-Tianjin-Hebei Region." *International Journal of Environmental Research and Public Health* 16 (14). <https://doi.org/10.3390/ijerph16142619>.

Reviewer #4 (Remarks to the Author):

I was asked to look in detail at the review and response to reviewer 3. Although I do agree it is important to cover positive impacts of afforestation, of which there are many, I think it's also important to step back and accept that authors can't be expected to quantify every aspect of every related question within a single paper. Moreover, the paper clearly states the context is the water balance trade-off between wetland extent and protection and afforestation. I also think it is very important to highlight potential water – vegetation trade-offs, with China in particular providing one of the largest (perhaps the largest?) natural experiments in the world in this respect, and I'm not sure it is reasonable to suggest such conclusions are harmful if all the caveats and assumptions are clear. In contrast to reviewer 3, I find the revised paper (and the authors response) quite nuanced across a complex topic and they navigate reasonably well a large number of uncertainties. That being said, I have a number of points and concerns that I feel the authors need to address for clarity of the key arguments and analysis:

1. Large numbers of area change given in abstract, very hard to interpret these numbers for the casual reader. I recommend making these a % relative to the initial areas.
2. Equation 3 in the methods is directly from Woodward et al.
3. Perhaps I have missed it, but I can't find anywhere how the authors are actually calculating PET. Is this a direct output of ORCHIDEE-Hillslope? Or is it an offline calculation made from a bunch of ORCHIDEE-Hillslope outputs? Does PET therefore evolve and have feedbacks as the land surface itself changes just as ET does? If so, I'm not sure what the annual average PET might mean if it is also trending with ET and Q.
4. The much higher sensitivity of the water yield / runoff in drier areas to land cover / forest

change has been covered in extensive detail in many papers, e.g. [Berghuijs, W. R., Larsen, J. R., van Emmerik, T. H. M., & Woods, R. A. (2017). A global assessment of runoff sensitivity to changes in precipitation, potential evaporation, and other factors. *Water Resources Research*, 53, 8475– 8486. <https://doi.org/10.1002/2017WR021593>]. The authors could consider referring to this previous finding already in the literature.

5. Looking at Figure 1d, it is clear that the majority of wetlands are trending to a decrease in area, however there is a non-negligible part of the distribution where wetland area is increasing with afforestation. These cases seem to be largely ignored in the rest of the analysis, but they provide an important counterpoint that deserves some attention, i.e. wetland loss isn't exclusive, and afforestation is clearly not always the key variable influencing wetland area (and also clearly not the trend in P). Alternatively, afforestation could still be the key variable, but the kind of afforestation (natural forest, monoculture, etc) being a nuance that is totally unknown. The authors might want to consider in the response analysis whether wetland area response is likely to be driven mostly by afforestation in all these cases (including both increasing and decreasing wetland trends). China has also experience landscape drainage and hydrological engineering on an unprecedented scale, and these human modifications could surely trump vegetation change influences in many cases. Another example, wetlands are by definition quite shallow, increased erosion and sediment loads from many sources (farmland erosion, urbanisation and construction, etc) could easily (and rapidly) infill the wetlands and completely change the bathymetry and drainage (and also cause the wetland to disappear). As it stands, there are interesting trends in the data that don't fit all the authors arguments and deserve some explanation.

6. My main concern is reproducibility and data availability. If one searches for ORCHIDEE-Hillslope, no documentation of the model setup or structure can be found. On the ORCHIDEE website, nothing about the hillslope version of the model is mentioned. There are profound assumptions on subsurface hydrology and how this connects to the surface to generate wetlands listed in the methods section, yet there does not appear to be any comprehensive source one can find that validates or provides basic sensitivity analysis to the many hydrological assumptions within ORCHIDEE-Hillslope. Moreover, how would anyone be able to replicate this analysis if the model, or it's description, are not even available? Since the model is not available, nor the model setups or data compilations used as inputs are also not available from the authors, all of which is incredibly disappointing. Moreover, writing "The GIEMS-2 inundated data are obtained from Catherine Prigent upon reasonable request" – basically means "if the author feels like it". Why not just make the data available? Thus, based on all these aspects the paper is not meeting any basic standards for reproducibility and data availability.

Response to the reviewers

To Reviewer #1

Reviewer #1 General comments

The revised manuscript has improved substantially, and now discusses the trade-offs of afforestation and wetlands in a much more balanced and informative way. The authors have addressed the majority of my previous concerns. My only remaining concern that I hope the authors could address as well, is the role of alternative potentially covarying drivers to changes in wetland exchange. Otherwise, most of my comments below are intended to help the authors increase the clarity and nuance of their writing.

[Response] We thank the reviewer for the positive feedback on our revised version and for the value comments and suggestions for improving our manuscript. Following the reviewer's comments and suggestions, we have 1) analyzed the role of alternative potentially covarying drivers on changes in wetland area, and added discussion about the other possible explanation aside from forestation for observed wetland loss/gain in the revised version; and 2) increased the clarity and nuance of the manuscript. Detailed point-by-point responses are listed following each comment/suggestion.

[Reviewer #1 General Comment 1]

Beginning of Abstract and Conclusions. Afforestation programmes in China and elsewhere are motivated by multiple purposes beside climate mitigation, which the authors also acknowledge (e.g., at L32: "In response to a national priority of protecting ecological services and land-system sustainability in the context of rapid economic development, China implemented a series of large-scale afforestation and forest protection programmes"). Nevertheless, the starting sentences in the Abstract and Conclusions strongly frame the primary motivation (and the perceived benefit) of afforestation programmes as "climate mitigation", and leaves the impression that the studied trade-off is between "climate" and "ecological" objectives. Thus, perhaps consider deleting "to solve the climate change problem" or adding other reasons, for a more nuanced and accurate description of the objectives of the afforestation programmes in China in the same sentence. Drawing the readers' attention to reasons such as "prevention of desertification and land degradation" also might enhance the understanding of the challenges to spatial optimization. I am aware that the authors acknowledge all the climatic, ecological, social development targets associated with both afforestation and wetlands, but those statements are a little more hidden, whereas the first sentences in the Abstract and Conclusions come off very strongly.

[Response] Thanks for the valuable suggestion. We have revised descriptions about the motivation of afforestation in China at the beginning of Abstract and Conclusions following the

reviewer's suggestions (Lines 14–15 and 249–251, also copied as below).

“For more than 20 years, China has been implementing programmes of widespread forestation to protect ecological services and land-system sustainability.”

“Over the next few decades, China will continue to implement a series of large-scale afforestation initiatives to protect ecological services and land-system sustainability and combat climate change^{19,39}.”

[Reviewer #1 General Comment 2]

L77 (i.e., section “Historical wetland change in response to afforestation”). While the other sections of the manuscript, better isolated the effect of afforestation, this section appears to claim to have isolated the “afforestation” effect (causation) despite essentially resting on the assumption that afforestation does not for example strongly co-vary with wetland drainage, irrigation expansion, or other water infrastructure projects.

[Response] In this section, we used the satellite-based wetland product, GIEMS-2 to show the observed wetland change. Over natural wetland grid cells (grid cells with a > 10% cover fraction of rice paddies were removed using the inventory-corrected HYDE v3.2 data set), we found that 65% of them experienced forest gains but decreasing wetland cover fraction (Fig. 1d). Consistent with the reviewer's concern, given that the observed wetland loss could result from the other drivers such as wetland drainage, irrigation expansion, or other water infrastructure projects which could co-vary with the change in forest area, we used a weak tone to suggest that the increasing forest cover fraction could be one possible explanation of the observed wetland loss, **“Such opposing trends for forest and wetland areas even in regions experiencing increased precipitation, suggest a possible negative impact of forestation on wetland areas in China”** (Lines 94–96). After that, the conceptual water-balance model, Budyko and the factorial simulations with and without forest change using ORCHIDEE-Hillslope can provide a robust causal attribution of the wetland area changes to afforestation.

[Reviewer #1 General Comment 3]

In my previous review, I was concerned that “alternative drivers” to wetland change were not sufficiently accounted for. By “alternative drivers”, I had in mind such land-use and water-use driven drivers (apologies that I did not articulate this even clearer in the previous exchange). For example, (Zhang et al. 2019) found “farm production, total aquatic products, and irrigated area” to be the top drivers of “wetland changes in the Beijing-Tianjin-Hebei Region”. The authors might want to address this in some way, for example, by adding layer(s) of trends in irrigation, agricultural areas, or other proxies of agricultural activities, and inspect the

covariation with forestation trends, and discuss the possibilities of such alternative explanations.

[Response] Thanks for the reviewer’s further explanation and suggestions. Following the reviewer’s suggestion, we obtained the statistics of annual human water withdrawal in China for three sectors including agriculture, industry, and domestic life from *China’s Water Resources Bulletin* for 2000–2016 (<http://www.mwr.gov.cn/sj/#tjgb>) and annual total completed investment for water infrastructure projects from *Statistic Bulletin on China Water Activities* for 2006–2016 (<http://www.mwr.gov.cn/sj/#tjgb>). The data for the wetland drainage in China are not available at grid or national scale. During the period 2000–2016, we found that the China’s agricultural, industrial, and domestic water use and the investment for water infrastructure projects show a significantly increasing trend of 1.3 billion m³ yr⁻², 1.4 billion m³ yr⁻², 1.6 billion m³ yr⁻², and 53.6 billion yuan yr⁻² ($p < 0.05$), respectively (Fig. R1), which could, accompanied with increasing forest coverage, contribute a decreasing wetland cover fraction in GIEMS-2. However, due to the lack of gridded data for human water use and water infrastructure projects, the further analyses of the impacts of forest change and the drivers related to human activities on wetland area at grid scale cannot be conducted. The HYDE v3.2 dataset can provide the information of change in irrigated agriculture area (Fig. R2), the impacts of increasing irrigated area on wetland area, however, is twofold: the increasing water use for irrigation could reduce water delivered to wetlands while the expanding irrigated area could increase the wetland area seen by GIEMS-2. The two opposite effects of irrigation on wetland change are hard to quantified.

Fig. R1. Temporal variations in human water withdrawal (a) and total completed investment for water infrastructure projects (b) in China from 2000 to 2016.

Fig. R2. Change in irrigated agriculture from HYDE v3.2 data set. **a**, Spatial pattern of trend in cover fraction of irrigated agriculture across wetland grid cells from 2000 to 2016. Hatching indicates the trend is statistically significant ($p < 0.05$). **b**, Temporal change of annual wetland area from GIEMS-2, forest area from China’s forest inventory data, and irrigated agriculture area from HYDE v3.2 data set from 2000 to 2016 across wetland grid cells.

Despite the uncertainties of inventory data for protected wetlands in China, one short cut to solve this problem is to analyze the impact of forest increase on wetland area only for grid cells located in basins containing protected wetlands, which could exclude the human disturbances including wetland drainage, water withdrawal, and water infrastructure projects to some degree. We found the similar results for protected wetland grid cells to Fig. 1d; there are 66% grid cells showing an increasing forest cover fraction but a decreasing wetland area and 67% among them experiencing an increasing annual precipitation (Fig. R3). This suggests that the historical forest gains in China do play a role in observed wetland loss, in addition to the land use and water use driven drivers.

Fig. R3. Trend in wetland fraction versus trend in forest cover fraction from 2000 to 2015 across wetland grid cells ($n = 562$) (**a**) and protected wetland grid cells ($n = 373$) (**b**). The colour of

each point shows the trend in annual precipitation (P) from GSWP3-W5E5. The inset at the bottom right indicates the probability density function of the trend in annual P across points in the fourth quadrant.

As a result, given the lack of gridded data for other alternative drivers and the substantial uncertainties in quantifying the impacts of change in forest area and other drivers on wetland area, we still kept our previous results in Fig. 1d, but added some discussion to provide a more comprehensive explanation of observed wetland loss from GIEMS-2 (Lines 94–98, also copied as below).

“Such opposing trends for forest and wetland areas even in regions experiencing increased precipitation, suggest a possible negative impact of forestation on wetland areas in China, despite the human disturbances including wetland drainage, irrigation expansion, or other water infrastructure projects could also have negative impacts²⁸⁻³⁰.”

[Reviewer #1 General Comment 4]

L241 (and Supplementary Table 2). I found it quite confusing that the contribution of forest change varied so greatly between (“accounting for about 23%, -92%, and 41% of wetland change from multi-model mean projections”) and that the term “contribution” is defined in a way that makes >100% contributions possible. If I understand the Supplementary Table 2 and the authors’ definitions correctly, the forest change effect on wetland extent is in fact quite consistent across the different scenarios, whereas the CC and eCO₂ effects can be erratic and both increase and decrease wetland extents. The authors might want to add an explanation of why CC and eCO₂ effects on wetland vary widely among the different scenarios. In addition, a different way to define “contribution”, or a different term to describe the S1-to-S0 ratio, might be helpful for the reader.

[Response] Thanks for the suggestion. The substantially different effects of CC and eCO₂ among different SSP scenarios are caused by the highly uncertain precipitation projected by different climate models under different scenarios. We have added an explanation for this at Lines 235–237 (also copied as below).

“Considerable disagreements in projected wetland change can be found due to the highly uncertain precipitation simulated by different climate models under different SSP scenarios (Supplementary Table 2).”

For the “contribution” term, we revised it as “**impact**” in the revised caption for Supplementary Table 2, and the contribution of forest was revised as “**relative impact of forest change to all three factors**” in turn, which could be more accessible to readers.

[Reviewer #1 General Comment 5]

Discussion/Conclusions. I appreciate the revised discussions and conclusions, which are much more nuanced. I find it a great addition that the authors added this at the end of the Conclusions: “The ecological trade-offs between wetlands and plantations should face the aspect of other ecosystem services such as high carbon sequestration, biodiversity, and cultural values. The implementation of future afforestation activities should also assess the feasibility of spatial optimization of afforestation goals and the priority of ecosystem services.”. The authors might want to add a few more sentences to explain what this means, and how challenging such trade-offs might or might not be to overcome. (For example, is climate mitigation through afforestation less efficient in the northern part less efficient due to albedo-effects anyway? Are the carbon stocks in wetlands large? Are the carbon stocks in wetlands well-accounted for? Etc.)

[Response] Thanks for the positive feedback on our revised discussions and conclusions. Following the reviewer’s suggestions, we added more sentences about the challenge in implementing the trade-offs of future forestation activities and wetland conservation (Lines 286–292, also copied as below).

“The ecological trade-offs between wetlands and plantations should also assess the effectiveness of forestation for carbon sequestration against wetlands⁴⁹, biophysical climate feedbacks such as decreasing albedo due to forestation in boreal regions⁵⁰, and other ecosystem services such as biodiversity conservation and cultural values¹⁷. The implementation of future forestation activities should also face the aspect of the feasibility of spatial optimization of afforestation goals and the priority of ecosystem services^{49,51,52}.”

[Reviewer #1 General Comment 6]

Supplementary Figure 11. Please update the in-figure subheadings S-A to S-D.

[Response] The figure has been updated.

[Reviewer #1 General Comment 7]

Terminology:

The term “afforestation” is normally reserved for forestation measures on historically non-forested land. In this manuscript, the term appears to be used to refer to both afforestation and reforestation. The authors might want to consider defining the term in the introduction to avoid confusion, or change the term to e.g., “forestation”, “tree cover increase”, or “afforestation and reforestation” etc.

[Response] Thanks for the kind reminder. The term “afforestation” in our study refers to both afforestation and reforestation. Following the reviewer’s suggestion, we have revised the term “afforestation” as “forestation” throughout the manuscript and added the definition of the term in the Introduction (Line 31).

[Reviewer #1 General Comment 8]

Could the authors also add a definition of wetlands? Perhaps by expanding the definition at L 85 and explaining what a wetland fraction indicates? (I understand that lakes and rice paddies are removed, but how about other artificial reservoirs for example?)

[Response] Thanks. In the section of “**Historical wetland change in response to forestation**”, we first calculated observed wetland change following the wetland definition from GIEMS-2 (wetlands in GIEMS-2 include open water, wetlands, and rice paddies, potentially exclude large lakes, rivers, and reservoirs; Prigent et al., 2020) (Fig. 1c), and then we removed the rice paddies using the HYDE v3.2 from GIEMS-2 to analyze the impacts of forestation on natural wetlands (Fig. 1d). The wetland extent after removing the rice paddies from GIEMS-2 and RFW (which has been explained in details in the section “**Observation-based wetland extent (flooded area)**” of Methods) is used to calibrate the parameters of our wetland model. Following the reviewer’s suggestions, we added the definition of wetlands for GIEMS-2, “**(including wetlands, open water, and rice paddies)**”, at Line 85 and added the reference to the Methods for calibration wetland products, “**(see wetland definition in Methods)**”, at Line 133.

References

- Hanasaki, N., Yoshikawa, S., Pokhrel, Y. & Kanae, S. A global hydrological simulation to specify the sources of water used by humans. *Hydrol. Earth Syst. Sci.* 22, 789-817, doi:10.5194/hess-22-789-2018 (2018).
- Klein Goldewijk, K., Beusen, A., Doelman, J. & Stehfest, E. Anthropogenic land use estimates for the Holocene – HYDE 3.2. *Earth Syst. Sci. Data* 9, 927-953, doi:10.5194/essd-9-927-2017 (2017).
- Prigent, C., Jimenez, C. & Bousquet, P. Satellite-Derived Global Surface Water Extent and Dynamics Over the Last 25 Years (GIEMS-2). *J. Geophys. Res.-Atmos.* **125**, e2019JD030711, doi:<https://doi.org/10.1029/2019JD030711> (2020).
- Zhang, Liyun, Quan Zhen, Min Cheng, and Zhiyun Ouyang. 2019. “The Main Drivers of Wetland Changes in the Beijing-Tianjin-Hebei Region.” *International Journal of Environmental Research and Public Health* 16 (14). <https://doi.org/10.3390/ijerph16142619>.

To Reviewer #4

Reviewer #4 General comments

I was asked to look in detail at the review and response to reviewer 3. Although I do agree it is important to cover positive impacts of afforestation, of which there are many, I think it's also important to step back and accept that authors can't be expected to quantify every aspect of every related question within a single paper. Moreover, the paper clearly states the context is the water balance trade-off between wetland extent and protection and afforestation. I also think it is very important to highlight potential water – vegetation trade-offs, with China in particular providing one of the largest (perhaps the largest?) natural experiments in the world in this respect, and I'm not sure it is reasonable to suggest such conclusions are harmful if all the caveats and assumptions are clear. In contrast to reviewer 3, I find the revised paper (and the authors response) quite nuanced across a complex topic and they navigate reasonably well a large number of uncertainties.

That being said, I have a number of points and concerns that I feel the authors need to address for clarity of the key arguments and analysis.

[Response] We thank the reviewer for supporting our study and the valuable comments and suggestions for improving this manuscript. Following the reviewer's comments and suggestions, we have 1) added more discussions and analyses to solve the reviewer's concerns; and 2) revised the Data availability and Code availability for data and model code publicly available to make our work reproducible. Detailed point-by-point responses are listed following each comment/suggestion.

[Reviewer #4 General Comment 1]

1. Large numbers of area change given in abstract, very hard to interpret these numbers for the casual reader. I recommend making these a % relative to the initial areas.

[Response] The percentage change relative to the initial areas has been added in the Abstract.

[Reviewer #4 General Comment 2]

2. Equation 3 in the methods is directly from Woodward et al.

[Response] We have added the citation of Woodward et al. (2014) for Equation (3).

[Reviewer #4 General Comment 3]

3. Perhaps I have missed it, but I can't find anywhere how the authors are actually calculating PET. Is this a direct output of ORCHIDEE-Hillslope? Or is it an offline calculation made from a bunch of ORCHIDEE-Hillslope outputs? Does PET therefore evolve and have feedbacks as

the land surface itself changes just as ET does? If so, I'm not sure what the annual average PET might mean if it is also trending with ET and Q.

[Response] PET in this study is output directly from ORCHIDEE-Hillslope, which is calculated using the unstressed surface-energy balance method (USEB; Barella-Ortiz et al., 2013). Therefore, the PET variations are only determined by several climatic variables including temperature, specific air humidity, wind speed, etc., but has no feedback as the land surface change. We have added the description “**The PET is output directly from ORCHIDEE-Hillslope.**” in the caption of Supplementary Fig. 4.

[Reviewer #4 General Comment 4]

4. The much higher sensitivity of the water yield / runoff in drier areas to land cover / forest change has been covered in extensive detail in many papers, e.g. [Berghuijs, W. R., Larsen, J. R., van Emmerik, T. H. M., & Woods, R. A. (2017). A global assessment of runoff sensitivity to changes in precipitation, potential evaporation, and other factors. Water Resources Research, 53, 8475– 8486. <https://doi.org/10.1002/2017WR021593>]. The authors could consider referring to this previous finding already in the literature.

[Response] Thanks. The paper suggested by the reviewer has been cited (Lines 630–632) in the part about the “**Budyko conceptual model**” of the Methods (Line 356).

[Reviewer #4 General Comment 5]

5. Looking at Figure 1d, it is clear that the majority of wetlands are trending to a decrease in area, however there is a non-negligible part of the distribution where wetland area is increasing with afforestation. These cases seem to be largely ignored in the rest of the analysis, but they provide an important counterpoint that deserves some attention, i.e. wetland loss isn't exclusive, and afforestation is clearly not always the key variable influencing wetland area (and also clearly not the trend in P). Alternatively, afforestation could still be the key variable, but the kind of afforestation (natural forest, monoculture, etc) being a nuance that is totally unknown. The authors might want to consider in the response analysis whether wetland area response is likely to be driven mostly by afforestation in all these cases (including both increasing and decreasing wetland trends). China has also experience landscape drainage and hydrological engineering on an unprecedented scale, and these human modifications could surely trump vegetation change influences in many cases. Another example, wetlands are by definition quite shallow, increased erosion and sediment loads from many sources (farmland erosion, urbanisation and construction, etc) could easily (and rapidly) infill the wetlands and completely change the bathymetry and drainage (and also cause the wetland to disappear). As

it stands, there are interesting trends in the data that don't fit all the authors arguments and deserve some explanation.

[Response] In Fig. 1d, there are 30% of grid cells experiencing afforestation and an increasing wetland cover fraction. Among these grid cells, there are more than 80% with an increasing annual precipitation. Hence, the wetland gains over these grid cells could be attributed more to increasing precipitation. While for those grid cells experiencing afforestation and wetland loss (~65%), over 60% of them have had an increasing annual precipitation. We thereby inferred that the wetland loss over these grid cells could be partly caused by afforestation, particularly for those with increasing precipitation. However, we agree with the reviewer that in addition to afforestation, the widespread human modifications such as landscape drainage, human water use, and hydrological engineering could also alter wetland areas. According to the statistics of annual human water withdrawal in China for three sectors including agriculture, industry, and domestic life from *China's Water Resources Bulletin* for 2000–2016 (<http://www.mwr.gov.cn/sj/#tjgb>) and annual total completed investment for water infrastructure projects from *Statistic Bulletin on China Water Activities* for 2006–2016 (<http://www.mwr.gov.cn/sj/#tjgb>), we found that the China's agricultural, industrial, and domestic water use and the investment for water infrastructure projects show a significantly increasing trend of 1.3 billion m³ yr⁻², 1.4 billion m³ yr⁻², 1.6 billion m³ yr⁻², and 53.6 billion yuan yr⁻² ($p < 0.05$), respectively (Fig. R1), which could, accompanied with increasing forest coverage, contribute a decreasing wetland cover fraction in GIEMS-2. However, due to the lack of gridded data for human water use and hydrological engineering, the further analyses of the impacts of forest change and the drivers related to human activities on wetland area at grid scale cannot be conducted. Given the lack of gridded data for these drivers and the substantial uncertainties in quantifying the impacts of change in forest area and other drivers on wetland area, we still kept our previous results in Fig. 1d, but added some discussion to provide a more comprehensive explanation of observed wetland loss from GIEMS-2 (Lines 94–98, also copied as below).

“Such opposing trends for forest and wetland areas even in regions experiencing increased precipitation, suggest a possible negative impact of forestation on wetland areas in China, despite the human disturbances including wetland drainage, irrigation expansion, or other water infrastructure projects could also have negative impacts²⁸⁻³⁰.”

Fig. R1. Temporal variations in human water withdrawal (a) and total completed investment for water infrastructure projects (b) in China from 2000 to 2016.

[Reviewer #4 General Comment 6]

6. My main concern is reproducibility and data availability. If one searches for ORCHIDEE-Hillslope, no documentation of the model setup or structure can be found. On the ORCHIDEE website, nothing about the hillslope version of the model is mentioned. There are profound assumptions on subsurface hydrology and how this connects to the surface to generate wetlands listed in the methods section, yet there does not appear to be any comprehensive source one can find that validates or provides basic sensitivity analysis to the many hydrological assumptions within ORCHIDEE-Hillslope. Moreover, how would anyone be able to replicate this analysis if the model, or its description, are not even available? Since the model is not available, nor the model setups or data compilations used as inputs are also not available from the authors, all of which is incredibly disappointing. Moreover, writing “The GIEMS-2 inundated data are obtained from Catherine Prigent upon reasonable request” – basically means “if the author feels like it”. Why not just make the data available? Thus, based on all these aspects the paper is not meeting any basic standards for reproducibility and data availability.

[Response] Thanks. Following the reviewer’s suggestions, we have revised the data and code availability and made the ORCHIDEE-Hillslope model code and GIEMS-2 data used in this study available (Lines 462–463 for Data availability and Lines 473–482 for Code availability, also copied as below). For ORCHIDEE-Hillslope, we created a new wiki page on the ORCHIDEE website (<https://forge.ipsl.jussieu.fr/orchidee/wiki/GroupActivities/CodeAvailabilityPublication/ORCHIDEE-Hillslope-r6515>), where the guidance to download the model code and corresponding references are provided. We cited the PhD thesis of Tootchi (2019) for the model, which fully describes all the equations and algorithms of ORCHIDEE-Hillslope. The pdf of Tootchi’s PhD

thesis has been put on the wiki page above, and can be directly downloaded by the link (https://www.metis.upmc.fr/~ducharne/documents/These_Tootchi_revised_11Sep2019.pdf).

The access for the input climate data and user guide to install and run the model are also given in the revised version. For GIEMS-2, we have uploaded the GIEMS-2 dataset used in this study over China from 2000 to 2015 in NetCDF format to Zenodo (<https://doi.org/10.5281/zenodo.5750962>). We are open to other suggestions of the reviewer may have.

“..... The GIEMS-2 dataset analyzed during the current study over China from 2000 to 2015 is available on Zenodo (<https://doi.org/10.5281/zenodo.5750962>)⁷⁶.”

“The ORCHIDEE-Hillslope model (r6515) code used in this study is open-source and distributed under the CeCILL (CEA CNRS INRIA Logiciel Libre) license. It can be available at <https://forge.ipsl.jussieu.fr/orchidee/wiki/GroupActivities/CodeAvailabilityPublication/ORCHIDEE-Hillslope-r6515>, with guidance to install and run the model at <https://forge.ipsl.jussieu.fr/orchidee/wiki/Documentation/UserGuide>. All ancillary data to run the model can be accessed upon reasonable request to the corresponding author. The code to simulate wetland area by TOPMODEL are publicly available on GitHub (https://github.com/yixixy/Wetland_simulation_by_TOPMODEL)⁷⁷. The processing MATLAB codes are available from the corresponding author upon reasonable request.”

References

- Barella-Ortiz, A., Polcher, J., Tuzet, A. & Laval, K. Potential evaporation estimation through an unstressed surface-energy balance and its sensitivity to climate change. *Hydrol. Earth Syst. Sci.* **17**, 4625-4639, doi:10.5194/hess-17-4625-2013 (2013).
- Tootchi, A. Development of a global wetland map and application to describe hillslope hydrology in the ORCHIDEE land surface model. Sorbonne Université, https://www.metis.upmc.fr/~ducharne/documents/These_Tootchi_revised_11Sep2019.pdf (2019).
- Woodward, C., Shulmeister, J., Larsen, J., Jacobsen, G. E. & Zawadzki, A. The hydrological legacy of deforestation on global wetlands. *Science* **346**, 844-847, doi:10.1126/science.1260510 (2014).

Reviewer comments, second round review -

Reviewer #1 (Remarks to the Author):

The authors have addressed my previous concerns in this revision. It is now nuanced and insightful, and offers a clear description of the limitations of the study.

(On a minor note, At L272-274, I found it unclear what the authors mean by "However, the feedback on precipitation has little alleviation or even aggravates the wetland loss risk to forestation in northeastern China with noticeable wetland coverage." In my understanding, the cited ref. 16 did not observe extensive forestation in northeastern China?)

Reviewer #4 (Remarks to the Author):

This paper has made great improvements and is indeed very comprehensive in its current form. The authors acknowledge factors other than afforestation could be impacting wetland loss, not in a terribly strong way, but this is their choice to make. This fits within the more general detection of water areas being converted to land (and vice versa), for which there was already a global analysis / overview (<https://www.nature.com/articles/nclimate3111>), and the authors may wish to cite. Nonetheless, this has two important implications for some key points:

- 1) The authors only estimate total wetland loss (1300 – 1500 km²) in the 2000 – 2016 period. However, a more complete picture would be to complement this with 'net' wetland loss, taking into account that some areas are increasing in wetland area. It might also be useful to ask how many wetlands basically have no change in area.
- 2) Currently, the model is calibrated to find a difference between two steady state scenarios, that as far as I can tell, are not accounting for the 30% of wetlands with observed increases. Thus, the prediction of an additional 1300 km² wetland loss (due to afforestation alone) is likely an overestimate if one considers both the 'net' effect as the observed data 2000 - 2016 shows (point 1), and the unknown contribution of engineering / sedimentation on wetland loss. This is a distinct point compared to uncertainty on future precipitation, which governs most of the prediction uncertainty covered by the authors.

Response to the reviewers

To Reviewer #1

Reviewer #1 comments

The authors have addressed my previous concerns in this revision. It is now nuanced and insightful, and offers a clear description of the limitations of the study.

(On a minor note, At L272-274, I found it unclear what the authors mean by “However, the feedback on precipitation has little alleviation or even aggravates the wetland loss risk to forestation in northeastern China with noticeable wetland coverage.” In my understanding, the cited ref. 16 did not observe extensive forestation in northeastern China?).

[Response] We thank the reviewer for the constructive and positive comments in all three reviews of this study. Following the reviewer’s comment in this round, we have removed the sentence in the manuscript.

To Reviewer #4

Reviewer #4 comments

This paper has made great improvements and is indeed very comprehensive in it’s current form. The authors acknowledge factors other than afforestation could be impacting wetland loss, not in a terribly strong way, but this is their choice to make. This fits within the more general detection of water areas being converted to land (and vice versa), for which there was already a global analysis / overview (<https://www.nature.com/articles/nclimate3111>), and the authors may wish to cite. Nonetheless, this has two important implications for some key points:

1) The authors only estimate total wetland loss (1300 – 1500 km²) in the 2000 – 2016 period. However, a more complete picture would be to complement this with ‘net’ wetland loss, taking into account that some areas are increasing in wetland area. It might also be useful to ask how many wetlands basically have no change in area.

2) Currently, the model is calibrated to find a difference between two steady state scenarios, that as far as I can tell, are not accounting for the 30% of wetlands with observed increases. Thus, the prediction of an additional 1300 km² wetland loss (due to afforestation alone) is likely an overestimate if one considers both the ‘net’ effect as the observed data 2000 - 2016 shows (point 1), and the unknown contribution of engineering / sedimentation on wetland loss. This is a distinct point compared to uncertainty on future precipitation, which governs most of the prediction uncertainty covered by the authors.

[Response] We thank the reviewer for the positive feedback on our revised version. The

suggested paper has been cited as ref. ²⁸ to support the detected conversion from water areas to land. For the reviewer's two comments, please find detailed point-to-point responses below.

- 1) The "1300–1500 km² wetland loss" in our study indicates the "net" wetland loss, not "total" wetland loss, due to tree planting. We have clarified the term "net wetland loss" throughout the manuscript.
- 2) Following our response to point 1), the wetland loss reported in our study indicates "net" wetland loss, the "real" wetland loss due to tree planting was thereby not overestimated regardless of the uncertainties of our predictions. As for the reviewer's concern about the contribution of engineering/sedimentation on wetland loss, it should be clarified that our two steady state scenarios were set to see the impacts of tree planting on wetland change, not to reproduce the observed wetland loss or gains. Except tree planting, engineering, sedimentation, human water consumption, and so on can also result in observed wetland change. These factors are major contributors to the uncertainties in predicting observed wetland change, but not to the uncertainties in predicting the impacts of tree planting on wetland change.